# Study of Symptom Severity and Adherence to Therapy of Myelofibrosis Patients Treated with Ruxolitinib

**DOI:** 10.3390/ph16070976

**Published:** 2023-07-07

**Authors:** Vera Stoeva, Guenka Petrova, Konstantin Mitov, Konstantin Tachkov

**Affiliations:** 1Specialized Hospital for Active Treatment of Hematological Diseases, 1756 Sofia, Bulgaria; 2Faculty of Pharmacy, Medical University of Sofia, 1756 Sofia, Bulgaria

**Keywords:** myelofibrosis, ruxolitinib, severity of symptoms, adherence to therapy

## Abstract

We aimed to explore symptom severity and adherence to therapy for patients with myelofibrosis treated with ruxolitinib in Bulgaria. It is a prospective, non-interventional study performed at the specialized hospital for active treatment of hematological diseases in Sofia during 2022–2023. Date of diagnosis, demographic characteristics, clinical indicators, ruxolitinib dose, and other data points were collected. Clinical indicators were assessed at baseline, in the middle, and at the end of observation. Severity of symptoms was measured with MPN-SAF TSS and adherence to therapy with the Morisky 4 questionnaire six times during the observation. The mean age of diagnosis was 58.5 years, with the average duration of disease of 3 years. Patients’ laboratory results were within physiological ranges, with spleen size experiencing a constant decrease. The average value for the severity of the symptoms per MPN-SAF TSS results decreased significantly, indicating better disease control. The average adherence to therapy did not change and remained high at around 9 points, except for one patient. In conclusion the treatment of myelofibrosis patients with ruxolitinib decreased symptom severity and spleen size. Patients were adherent to the therapy over the observed period, but as treatment duration increases, the risk of adherence decreases.

## 1. Introduction

Rare diseases are an object of intensive study, especially after the advancement of their therapy with the creation of many new targeted therapies, but little is known about the adherence of patients to these new molecules. Myelofibrosis (MF) is a rare, hematologic malignancy that is pathologically characterized by bone marrow fibrosis, extramedullary hematopoiesis, and an overactive Janus kinase inhibitor signal transducer and activator of transcription protein (JAK-STAT) pathway. Clinically, the disease is characterized by splenomegaly, cytopenias, and constitutional symptoms, including fever, night sweats, and weight loss [1]. These constitutional symptoms can be debilitating, compromise the quality of life in MF patients, and reduce adherence to therapy [2]. 

Current pharmacotherapy of myelofibrosis in adults worldwide is based on two medicines authorized for use by the Food and Drug Administration (FDA) and the European Medicines Agency (EMA), the Janus kinase inhibitors (JAKi), ruxolitinib and fedratinib [3,4,5,6], with recent studies indicating an increase in the median survival of patients in the last decade from 48 to 63 months [7]. Ruxolitinib is a Janus kinase inhibitor used to treat various types of myelofibrosis and polycythemia vera in patients who have not responded to or cannot tolerate hydroxyurea and to treat graft-versus-host disease in cases that are refractory to steroid treatment. The Janus kinase (JAK) family of protein tyrosine kinases comprises JAK1, JAK2, JAK3, and non-receptor tyrosine kinase 2 (TYK2) [8]. JAKs play a pivotal role in intracellular signaling pathways of various cytokines and growth factors essential to hematopoiesis, such as interleukin, erythropoietin, and thrombopoietin [9]. JAKs have diverse functions. JAK1 and JAK3 promote lymphocyte differentiation, survival, and function, while JAK2 promotes signal transduction of erythropoietin and thrombopoietin [8,10]. JAKs are in close proximity to the cytokine and growth factor receptor’s cytoplasmic region. Upon binding of cytokines and growth factors, JAKs are activated, undergoing cross-phosphorylation and tyrosine phosphorylation. This process also reveals selective binding sites for STATs, which are DNA-binding proteins that also bind to the cytoplasmic region of cytokine or growth factor receptors. Activated JAKs and STATs translocate to the nucleus as transcription factors to regulate gene expression of pro-inflammatory cytokines, such as IL-6, IL-10, and nuclear factor κB (NF-κB) [11]. They also activate downstream pathways that promote erythroid, myeloid, and megakaryocytic development [8]. By inhibiting JAK1 and JAK2, ruxolitinib works to block the dysregulated cell-signaling pathways and prevents abnormal blood cell proliferation [12].

As an organic compound, ruxolitinib belongs to the pyrrolo [2,3-d] pyrimidines, containing an aromatic heteropolycyclic isomer of pyrrolopyrimidine with the 3 ring nitrogen atoms at the 1-, 5-, and 7-positions (Figure 1).

Nonadherence to therapy is recognized as one of the most important and costly problems for global health in the 21st century. According to a European Commission report, non-adherence to therapy is responsible for 194,500 deaths and costs the EU an estimated EUR 125 billion annually. As such, measuring adherence can be extremely useful to better understand patient behavior and outcomes [13]. The World Health Organization (WHO) describes groups of factors that might negatively influence adherence to therapy, and among them, are the severity and duration of the disease, adverse events, or intolerance to the therapy [14].

Adherence to therapy of myelofibrosis patients in real life practice has not been studied until now. Most studies measure the adherence in clinical trials, but in real life, patients might experience many different obstacles preventing them from following the recommendations. Therefore, studying the adherence to therapy of patients in a real-life setting is important to gain information for the probable risk of deterioration in the patient’s status. 

MF treatment guidelines recommend assessing the severity of the patients’ symptoms during clinical visits. The Myeloproliferative Neoplasm Symptom Score from the Total Symptom Score (MPN-SAF TSS) is a standardized system that rates 10 specific symptoms associated with MPN on a scale of 1–10. A study that aims to provide up-to-date guidelines and practice recommendations and highlights barriers to adherence in the long-term management of chronic myeloproliferative neoplasms (MPN), examining current non-curative drug therapy for MF that prolongs survival [15]. Identifying potentially modifiable barriers to treatment, they show that good adherence allows for the implementation of an intervention in a timely manner, improving clinical results and overall adherence. Authors consider that the long duration and disease course may contribute to the high risk of nonadherence in this group of patients. Poor adherence to long-term therapies seriously compromises treatment effectiveness. The MPN-SAF TSS questionnaire is designed to evaluate in a coherent way the influence of the severity of symptoms on the disease progression, and many factors previously described by the WHO are included in its construction. Studying the influence of the severity of symptoms on the adherence to therapy is important and allows evaluating these two factors as dependent. Such studies are scarce in the area of rare disease and orphan medicines. 

In this study we aimed to explore symptom severity and adherence to therapy for patients with myelofibrosis treated with ruxolitinib in an ambulatory department of tertiary care in Bulgaria.

The study explores the demographic characteristics of the observed patients in its first part, then the changes in the MPN SAF score during the therapy in the second part, and at the end, the changes in adherence to therapy during the observed period. Statistical correlation between the observed indicators was thoroughly evaluated to answer the question whether there are any statistical dependencies between the observed variables. 

## 2. Results

### 2.1. Demographics, Severity of Symptoms, and Adherence to Therapy

Table 1 presents the demographic data for the studied patients after initiation of ruxolitinib therapy.

They were proportionally distributed by gender, with a mean age at diagnosis of 58.5 years, which is typical of adult-onset diseases. The duration of the disease is an average of 3 years, but there is one patient with an 8-year long therapy. All observed patients during the follow-up period remained alive.

From the clinical point of view, the minimum allowed dose is 5 mg twice daily, but patients are usually assigned higher doses due to diagnosis in the advance stage. If patients develop intolerance or adverse events, the dose is decreased. Therefore, the main reason for doses changes is patient safety.

Table 2 presents the changes in clinical indicators during the follow-up period. The time point is at the beginning of the observation, in the middle, and at the end of the observation period for each patient. Despite inter-patient variation, the blood results remain within the physiological range of their values. The most important indicator for disease control, spleen size, showed a tendency to decrease, indicating the effectiveness of the therapy. Spleen size is the major indicator for disease control. The patients have variations in the initial spleen sizes measured in centimeters that might explain the small SD in comparison with the other clinical indicators. We observed minor but noticeable reductions in spleen size, indicating effectiveness of therapy; however, SD values are typical for such a low number of patients. The observed increases in leukocytes, hemoglobin, and platelet levels may be explained both by a change in the disease and by the treatment being carried out. According to clinical guidelines, clinical tests are performed every three months to establish their range. If the range is within the physiological intervals, no other measures are recommended.

The average value for the severity of the symptoms, measured with MPN-SAF TSS, decreased significantly over the follow-up period, indicating that patients experienced better disease control (Table 3). The average adherence to therapy did not change and remained high throughout, near the maximum value of 10 points. Morisky test response values above 8 are considered to be good adherence, which also corresponds to the behavior of patients with myelofibrosis (Table 3).

At the individual level, values of MPN-SAF-TSS decreased in general (Figure 2). Of all 21 patients for whom all six measurements were available, 12 reported lower indicator values, with the remaining reports still showing a general trend toward decreasing, despite variation.

Inter-patient variability of adherence, as measured by the Morisky test, is shown in Figure 3. In only one of the patients, a lower adherence was recorded in two of the measurements (third and fifth). We checked the individual patient data for the 14th patient and found out that the patient is a female at the age of 62 with primary MF, homozygotes for JAK2 V617F mutation and with 8 years length of the disease. She had a decrease in spleen size from 18 to 16 mm and clinical indicators within the physiological range. These results confirm that the long-lasting therapy negatively influences the adherence [14].

### 2.2. Statistical Analysis

Percentage of males in the study were 52.4% and percentage of females were 47.6%, with no statistical difference between their numbers. The Mann–Whitney test showed that there was no statistically significant difference between the average age of both gender groups (*p* = 0.2197) as well as in distribution of patients according to their JAK status (*p* = 0.5174). In contrast, we found that there is a difference in the length of the disease between the male and female groups (*p* = 0.0241). The length of disease in the male group was 2.7 years longer than female. This could be explained with the fact that the myelofibrosis in general is more common for males, and they probably were diagnosed earlier. 

We further explore the changes in the clinical indicators during the observed period. A significant reduction was found in the averages of the leukocytes between the three measurements from 12 to 8 g/L. The reduction between the first and last measurements was statistically significant with *p* < 0.05. 

The average values of hemoglobin are also decreased from 118.5 to 109.5 g/L in a statistically significant manner between the first and last measurements. We also observed a strong and significant reduction of hemoglobin levels between the second and third measurements (r = 0.952; *p* = 0.003). This trend analysis allows physicians to predict the changes in the level of hemoglobin for the next period. 

The thrombocytes average level decreased from 355 to 281 g/L non significantly (*p* > 0.05), but it correlated strongly and positively between the second and third measurements (r = 0.881; *p* = 0.0039), which might have prognostic value for physicians.

The values of lactate dehydrogenase (LDH) were similar, with average values decreasing from 506.5 to 404 U/L. A strong and statistically significant correlation was also found between value changes (r = 0.905; *p* = 0.0020). 

All clinical indicators remained within their reference intervals. 

The average size of the spleen decreased significantly from 21.8 to 19.3 cm (*p* < 0.0001). Spleen size is one of the most important prognostic indicators for the progress of the disease and effectiveness of therapy. The fact that it is decreased was a sign of the therapy success.

The statistical relationships between demographic and clinical characteristics of patients, as well as adherence to therapy and severity of symptoms were further investigated.

Measurement of severity of symptoms with MPN-SAF TSS showed a statistically significant difference in the reduction of mean values (*p* = 0.02216). Patients reported a steady reduction in values for the whole period of observation. This reduction is clinically relevant and statistically significant as the MPN-SAF TSS decrease indicated an improvement in clinical condition.

There was no statistically significant difference in the mean adherence to therapy measured with Morisky (*p* = 0.7151). Throughout all six inquiries, patient-reported mean values remained steady, with nonsignificant changes. We observed minimally improving adherence throughout the first five observations, with the value decreasing minimally as well in the sixth observation of follow-up. A decrease in mean values was indicative of a decrease in adherence and was a negative result. As in other studies, with long-lasting therapy, adherence declines, which we were unfortunately unable to follow for longer in our cohort.

Factors that have been studied and that may influence adherence and severity of symptoms are gender, JAK status, disease duration, and age. 

No statistically significant correlation was found between the Morisky scale, MFN-SAF TSS values, and gender (*p* > 0.05). Therefore, gender did not influence adherence to therapy and severity of symptoms (Table 4). JAK status also did not affect adherence and severity of symptoms.

Disease duration correlated with severity of symptoms in the third observation (r = 0.555), with adherence to therapy and severity of symptoms in the fourth observation (r = −0.397 and r = 0.527, respectively) (Table 5). The negative correlation between disease duration and adherence indicated that as treatment progressed, adherence decreased.

The age of the patients at the beginning of the follow-up did not affect the studied indicators, but on the sixth observation, a negative and statistically significant correlation appeared between age and adherence to therapy (r = 0.605, *p* < 0.05). This means that older patients have a worse adherence to therapy.

## 3. Discussion

In its comprehensive work on the adherence to long-term therapies, the WHO sets the scene for analyzing the influence of the poor adherence on the treatment of chronic diseases [14]. The growth of poor adherence is pointed out as disturbing because 50% of the patients with chronic diseases in developed countries do not follow the recommendations of physicians and pharmacists. Poor adherence leads to increasing cost of therapy, negative outcomes, and compromise patients’ safety. This is especially true for long-term therapies because their effectiveness might be severely compromised. Studying and analyzing the determinants of adherence in severely ill patients as is the case of MF allows enlightening the overall performance of the treating physicians and patients. As pointed out by WHO, health outcomes cannot be accurately assessed if they are measured predominantly by resource utilization and not by the effectiveness of interventions [14]. In this sense, simultaneously studying adherence and severity of the diseases provide information for the probably risks of compromising the therapy and increasing the severity and treatment cost. It is also very important for physicians to take adequate measures to enhance the adherence and to prevent any further patient deterioration. 

In this pilot study, we explored the changes in the clinical indicators, severity of symptoms, and adherence to therapy in a cohort of 21 patients with a rare disease, such as myelofibrosis, treated with ruxolitinib at the ambulatory department of tertiary care. To the best of our knowledge, there is no such study performed in the country. Patients are diagnosed in SHATHD, or they are diagnosed in other hospitals and then referred to SHATHD for treatment. Ruxolitinib is the only specific treatment option for patients with MF in Bulgaria. A hematology committee makes the decision if the patient is eligible to be treated with ruxolitinib. When the patient starts treatment, he needs to visit the outpatient department regularly every month. He has regular blood testing and, if necessary, the attending physician can adjust drug dose at his discretion. Every six months the response is re-evaluated. We use MPN-SAF TSS and spleen size measurement for the evaluation. Ultrasound and CT scan are the methods for spleen size evaluation. The treatment continues until disease progression or intolerance to treatment. If treatment change is indicated, it is discussed with the committee.

Fedratinib is not available on the national market and is not included in the study. The main limitation of the JAKi for MF patients are the hematological adverse events, mainly anemia and trombocitopenia. There are new JAKi as well as other molecules in the development stage aiming to overcome those adverse events [15].

Although small, the cohort of 21 patients is representative of the population attending the hospital—67 MF patients treated in the hospital. In our study we found that ruxolitinib improves clinical presentation of the patients and decreases spleen size. In addition, the changes in the severity of the disease score, evaluated with MPN-SAF, were statistically significant. We also found that older patients have a worse adherence to therapy as well as patients on long-lasting therapies. Although inconclusive, we observed a gradual decrease in adherence when the therapy had been applied for multiple years, with one patient showing some non-adherent periods. Valuable information for the physicians is the prognostic nature of some of the established correlations as the hemoglobin, thrombocytes values, and spleen size change. As was pointed out, the changes in the blood tests are indicative for the physicians for any hematological adverse events, but it is not disturbing if they remain within the physiological range [16]. 

The Morisky Medication Adherence Scale is a widely used questionnaire to indirectly assess patient adherence to medication therapy [17,18,19,20]. ROMEI is a prospective observational study focused on the treatment of MF patients in real-world settings with ruxolitinib [13]. The study enrolled 215 patients and included a prospective assessment of the medication adherence as a secondary endpoint. Of the 215 patients enrolled, 188 were assessed for medication adherence. It appears that patients with high adherence in week 4 were more likely to maintain the same level over time. No correlation was found between the status of the spleen and the level of adherence demonstrated (data available at the meeting). This analysis shows that a percentage of patients (ranging from 25% to 40%) belong to low–moderate levels of adherence during treatment. Surprisingly, overall, approximately one-third of patients may be exposed to suboptimal treatment over time, and clinicians likely underestimate the number of these patients in clinical practice. 

The continuation of the same study suggests that one-third of patients receiving ruxolitinib may be undertreated due to nonadherence, potentially undermining disease control, and indicates a need for better interventions targeting nonadherence to oral therapies [21]. Although no correlation was found between the level of adherence and the response of the spleen to treatment, probably due to the small number of patients evaluated for both variables, knowledge about the attitudes of patients to therapies, especially for long-term and chronic treatment, remains crucial from clinical and pharmacoeconomic studies. Standardized methods for assessing patient adherence, such as the MMAS-8, should be implemented in prospective well-designed clinical trials and could be a valuable tool to guide clinicians in daily practice. Furthermore, real-world studies of adherence complement secondary endpoints used in clinical trials.

Similar to our study is that of Palandri, which measures the relationships between the adherence, spleen answer, and distress [22]. It confirms the results from our study that duration of therapy negatively affects adherence and adds information about the negative influence of the high distress. The authors highlight that the association between low adherence and spleen response should be evaluated in real-life practice because it may improve MF therapy and prognosis. We fully agree with such a recommendation towards the physicians. Currently, there is no requirement to measure the adherence in real-life practice, but the results from our study and that of Palandri are in favor of introducing such an instrument. 

We did not find any other study exploring the relationship between the MPN-SAF and adherence despite the fact that other studies focusing on the quality of life of MF patients explore its dependency on the severity of the disease measured with MPN-SAF [23].

Limitations of our study are in the fact that it includes patients from only one clinic, but we must note that the total number of MF patients in Bulgaria is approximately 280 and out of them, nearly 70 (25%) are treated at SHATHD in Sofia. Although all patients were contacted for the purposes of the study, not all of them agreed to participate, probably due to the lack of time to answer every month or other reasons. This might influence the validity of the results for all MF patients. After the introduction of ruxolitinib in 2016, it began to be prescribed to all MF patients with manifested symptoms, and this is the first national study focusing on the characteristics of those patients, progress of the diseases, and adherence to therapy. 

## 4. Materials and Methods

### 4.1. Design of the Study

It is a prospective, non-interventional, observational study performed at the specialized hospital for active treatment of hematological diseases (SHATHD) in Sofia, Bulgaria during the period January 2022–February 2023. The SHATHD is the largest center in the country, currently treating more than one third of all patients with MF in Bulgaria. 

Out of 67 patients treated with ruxolitinib in the hospital, 21 were randomly selected and agreed to participate in the study. The sample size was limited to the number of patients that agreed to participate. Although all patients were contacted, not all were eager to answer the questionnaires every month. The patients were consecutively included in the follow-up by the attending physician, and the data on the demographic characteristics, clinical indicators, severity of symptoms, and adherence to therapy were collected and systematized. Inclusion criteria were any type of MF, regular visits every third month, treatment in the SHATHD, being on ruxolitinib, and willingness to participate in the observation and answer questionnaires. Exclusion criteria were non-MF patients, not treated in SHATHD, or unwillingness to participate. 

The following characteristics were collected in the beginning of observation: demographics, age at diagnosis, age at start of treatment with ruxolitinib, gender, type of myelofibrosis (primary myelofibrosis (PMF), post-polycythemia vera MF (post-PV MF), post-essential thrombocythemia MF (post-ET MF), DIPSS-calculated risk score (low, intermediate-1, intermediate-2, high risk), JAK2 mutation status (positive homozygous, positive heterozygous, or negative)), and other mutations were collected. 

From the clinical indicators, ruxolitinib dose, leukocytes, hemoglobin, platelets, lactate dehydrogenase (LDH), and spleen size were tracked. Clinical indicators were assessed at baseline, in the middle, and at the end of observation during the six months of the study, with blood tests carried at baseline, after 3 and 6 months. The clinical tests were performed at the hospital after collecting blood samples. Collection of blood samples is a routine practice during the follow up examination of MF patients and is included in the treatment package. Spleen size is measured via ultrasound and CT scan. 

Severity of symptoms was measured with the Myeloproliferative Neoplasm Symptom Assessment Form Total Symptom Score (MPN-SAF TSS). The MPN-SAF TSS questions focus on fatigue, concentration problems, early satiety, inactivity, night sweats, itching, bone pain, abdominal discomfort, weight loss, and fever. The decrease in the values pointing to improvement of the symptoms.

Adherence to therapy was measured with the Morisky 4 scale questionnaire, which assesses adherence on a scale of 0–10 points, with 10 points being complete adherence to therapy, and below 8 points non-adherence. 

Severity of symptoms, and adherence to therapy were monitored 6 times during the observation period (once every month), with patient surveys being conducted in the ambulatory department by the attending physician. Patients answer the questions in front of the physician, and if any issues appeared, they were solved on time. The measurement of the severity of symptoms with the Myeloproliferative Neoplasm Symptom Assessment Form Total Symptom Score (MPN-SAF TSS) is a routine practice in the clinic, but the measurement of adherence to therapy was performed for the first time. 

The ethical committee of the hospital approved the study (Order 3–31), and patients sign informed consent. 

### 4.2. Statistical Analysis

Descriptive statistic, Mann–Freidman test, and Pearson correlation analysis were performed with Medcalc ver. 13. Descriptive statistics were applied towards the demographic data, results from the clinical tests, MPN-SAF TSS, and adherence tests. Mann–Friedman test as non-parametric analysis was applied to test the statistical significance in the changes between different measurements of observed variables. Pearson correlation analysis was performed to test the probable correlation between the patients’ characteristics, severity of symptoms, and adherence to therapy. 

## 5. Conclusions

The treatment of myelofibrosis patients with ruxolitinib decreases the symptom severity and spleen size. Patients were adherent to the therapy over the observed period, but as treatment duration increases, the risk of adherence decreases. Simultaneously studying the adherence and severity of the diseases provides information for the probable risks of compromising the therapy, increasing the severity, and worsening the results. Adding the adherence evaluation in the armamentarium of the overall management of patients with myelofibrosis will enable physicians to better understand the results of the long-term therapy and to react in case of poor patient performance. 

## Figures and Tables

**Figure 1 pharmaceuticals-16-00976-f001:**
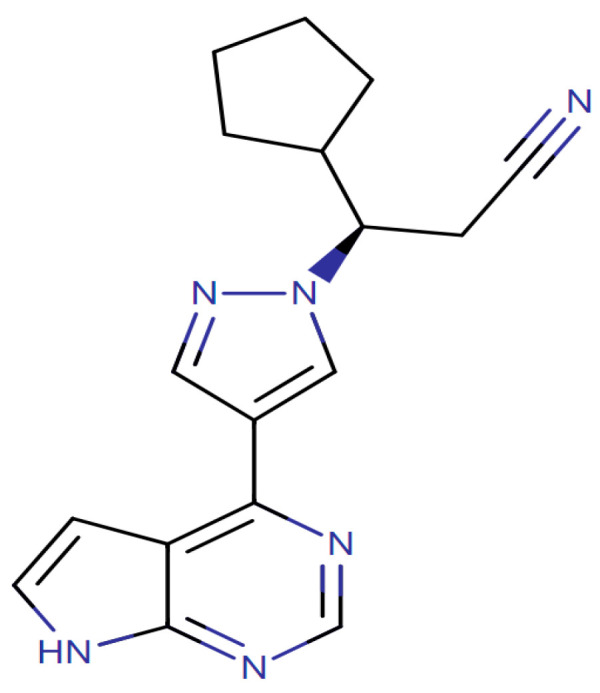
Chemical structure of ruxolitinib.

**Figure 2 pharmaceuticals-16-00976-f002:**
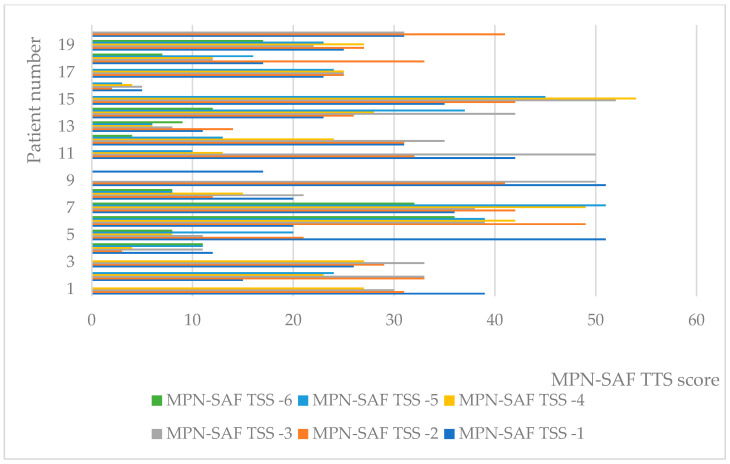
Individual values of MPN-SAF TTS.

**Figure 3 pharmaceuticals-16-00976-f003:**
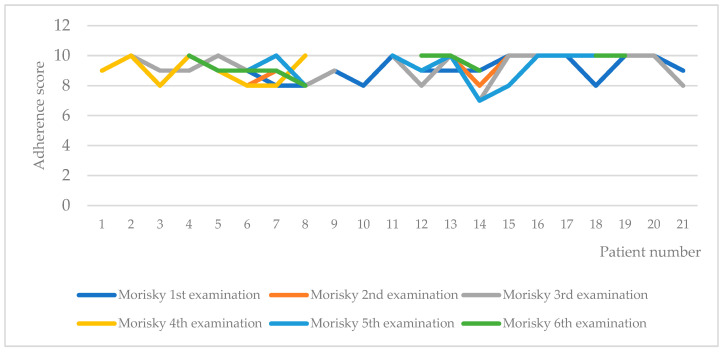
Adherence to therapy according to Morisky test.

**Table 1 pharmaceuticals-16-00976-t001:** Demographic characteristics of included patients on ruxolitinib therapy.

Characteristic	N	Average (SD)
Total number	21	
Female	10	48%
Male	11	52%
Age at the moment of observation	from 39 to 76 years	61.2 (8.4)
Age at diagnosis	from 34 to 70 years	58.49 (9.13)
Time from diagnosis of the disease	from 0.7 to 8 years	3.16 (1.88)
Type of myelofibrosis		
PMF	10	48%
Post-PV MF	9	42%
Post-ET MF	2	10%
Risk group		
Intermediate-1 risk	5	24%
Intermediate-2 risk	5	24%
Unknown	2	10%
High risk	3	14%
Low risk	6	28%
JAK2 V617F mutation status		
Homozygote mutation	6	28%
Heterozygote mutation	8	38%
Negative for mutation	4	20%
No information	3	14%
Ruxolitinib dose		
5 mg 2 × daily	3	14%
10 mg 2 × daily	6	30%
15 mg 2 × daily	8	38%
20 mg 2 × daily	3	14%

**Table 2 pharmaceuticals-16-00976-t002:** Changes in the clinical indicators of the included patients.

Indicator	Second Measurement–Average (SD)	Third Measurement–Average (SD)	
Leukocytes (G/L)	14.38 (7.9)	20.02 (16.16)	18.57 (3.26)
Hemoglobin (g/L)	106.92 (24.2)	117.82 (23.44)	110.38 (17.13)
Thrombocytes (G/L)	328.3 (150.1)	354.27 (165.26)	260.75 (93.25)
LDH (U/L)	688.38 (337.67)	751.9 (547.34)	626.25 (414.63)
Spleen size (cm)	21.8 (2.88)	19.5 (3)	19.33 (2.2)

**Table 3 pharmaceuticals-16-00976-t003:** Results from the evaluation of the severity symptoms and adherence to therapy.

Instrument	1st Inquiry Average (SD)	2nd Inquiry Average (SD)	3rd Inquiry Average (SD)	4th Inquiry Average (SD)	5th Inquiry Average (SD)	6th Inquiry Average (SD)
MPN-SAF TSS	26.7 (11.06)	28.8 (12.2)	26.09 (13.9)	20.4 (12.8)	18.3 (12.9)	14.4 (8.36)
Morisky	9.1 (0.7)	8.8 (1.3)	9.5 (0.9)	9.1 (0.8)	9.3 (0.8)	9.3 (0.7)

**Table 4 pharmaceuticals-16-00976-t004:** Pearson correlation between Morisky, MPN-SAF TSS scale, and gender.

Variable	GENDER = 1		GENDER = 2		
	Median	Average Rank	Median	Average Rank	P
MOR	9.0000	12.0500	9.0000	10.0455	0.4300
MPN_SAF_TSS	25.5000	11.3000	18.5000	9.7000	0.5446

**Table 5 pharmaceuticals-16-00976-t005:** Correlation between Morisky, MPN-SAF TSS scale, and disease duration.

Disease Duration (DD)		Morisky 3rd observation	MPN_SAF_TSS_3rd observation
DD	Correlation coefficient Significance Level P n	−0.153 0.5208 20	0.555 0.0137 19
Disease duration		Morisky 4th observation	MPN_SAF_TSS_4th observation
DD	Correlation coefficient Significance Level P n	−0.397 0.1024 18	0.527 0.0298 17

## Data Availability

The data that support the findings of this study are available from the corresponding author (G.P.) upon reasonable request.

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
