# Peer review of "Study of Symptom Severity and Adherence to Therapy of Myelofibrosis Patients Treated with Ruxolitinib"

_pharmaceuticals, 2023, doi:10.3390/ph16070976_

Round 1

Reviewer 1 Report

This pilot one-center study (21 participants) investigates symptoms' severity and adherence to therapy for patients with myelofibrosis treated with ruxolitinib in Bulgaria.

Even though the results sound interesting, it is highly advisable to introduce several suggestions to improve the quality of the study. Namely, inclusion and exclusion criteria should be more precisely defined as well as sample size estimation and power analysis for clinical research studies should be performed. Insufficient number of references has been detected. Additionally, discussion should be more thoroughly written.

Author Response

Dear reviewer,

thank you for your suggestions. We added comments for the sample selection, inclusions and exclusion criteis, as well as added additional references and discusiion. All changes are marked in track chages in the text.

Reviewer 2 Report

Vera Stoeva and colleagues studied symptom severity and treatment adherence in patients with myelofibrosis treated with ruxolitinib. The results found that ruxolitinib treatment of patients with myelofibrosis reduced symptom severity and spleen size. This study is interesting, but has some problems:

1.      Generally, statistics and results are together listed, why is the statistical analysis of the article only listed separately?

2.      What is the reason for the two "No information" for ruxolitinib dose in Table 1?

3.      Is there any correlation between the efficacy of ruxolitinib and ruxolitinib dose?

4.      What are the time points for each of the three measurements in Table 2?

5.      Several data in the First measurement-average column in Table 2 are lower than the Second measurement-average column (Thrombocytes and LDH); does the elevated Second measurement indicate poor efficacy?

6.      In Table 2, why is the average/SD of spleen size significantly larger compared with the average/SD of other indicators?

7.      Some numbers in the statistical analysis section do not coincide with those in the tables, where the table is the average and the narrative is the median.

8.      175 lines, isn't the value of spleen size from 23 to 29 cm increasing?

9.      135-136 lines, "At the individual level, values of MPN-SAF-TSS decreased in general (Figure 1). Of all 19 patients for whom all six measurements were available", but didn't the article count 21 patients?

Author Response

Dear reviwer, thank you for your valuabe comments. All changes are made in the text in track changes as well as in the attached file are addedd answers to your queries.

Reviewer 3 Report

Review on the manuscript

Manuscript ID : pharmaceuticals-2429210

Manuscript Title: Study of the symptoms’ severity and adherence to therapy of myelofibrosis patients treated with ruxolitinib

In this study the authors aimed to explore symptoms severity and adherence to therapy for  patients with myelofibrosis when treated with ruxolitinib in a tertiary care department in Bulgaria.  It explored the demographic characteristics of the  patients,  the changes in the MPN SAF score during the therapy and at the end the changes in adherence to therapy during the observed period. Statistical correlation between the observed indicators was evaluated.

In my opinion the paper may be accepted for publication with following revisions.

1.      Out of 67 patients treated with ruxolitinib in the hospital 21 were randomly selected 74 and agreed to participate in the study. There is a limitation in the study with respect to the number of patients and that too from only one clinic.  How to address it.

2.      The structure of Ruxolitinib and Fedratinib may be given.

3.      The mode of action of Ruxolitinib and  Fedratinib may be also included.

4.      Why only two drugs are available in the market.

 Only minor revision is required. 

Author Response

(The authors gave the same response as above.)

Reviewer 4 Report

The manuscript entitled "Study of the symptoms' severity and adherence to therapy of myelofibrosis patients treated with ruxolitinib" presents a clinical study conducted in Bulgaria that aims to explore the impact of ruxolitinib on the severity of symptoms and adherence to therapy among patients with myelofibrosis. While the topic of this study is interesting and addresses an important concern, several significant flaws throughout the manuscript need to be addressed before considering it suitable for publication.

One of the positive aspects of the manuscript is that it attempts to investigate the impact of ruxolitinib on the severity of symptoms in myelofibrosis patients. This focus on a specific drug and its effects is valuable for understanding the potential benefits of this treatment option. Additionally, the topic of adherence to therapy in patients with myelofibrosis is an important area to explore, as non-adherence can significantly impact treatment outcomes.

However, there are numerous shortcomings that undermine the manuscript's quality. Firstly, the introduction, although grounded in statistics, needs more clarity in its presentation of the study rationale and procedures. This lack of cohesiveness can make it challenging for readers to follow the author's intentions and the study's objectives. It is crucial to provide a clear and logical flow of information to engage readers effectively.

Furthermore, the manuscript suffers from a lack of relevant references cited in various sections. Referencing is essential for supporting the arguments made and providing a robust scientific foundation. By incorporating relevant citations, the manuscript would benefit from increased credibility and contextualization within the existing literature.

The Material and Methods section is also inadequate, as it lacks crucial experimental details and essential pieces of information. To ensure reproducibility and transparency, the authors should comprehensively describe the study design, patient selection criteria, data collection methods, and statistical analyses employed. This level of detail is essential for other researchers to assess the study's validity and replicate the findings.

Moreover, the manuscript fails to provide logical explanations about the type of study conducted to explore adherence and severity to therapy in patients with myelofibrosis treated with ruxolitinib. It is essential to clearly outline the study design (e.g., observational, experimental, etc.) to provide readers with a solid understanding of the methodology employed and the limitations associated with the chosen approach.

Additionally, the study appears to be a simple observational study that only presents the demographic characteristics of the observed patients. This limited scope prevents readers from fully grasping the essence of the report and understanding the broader implications of the findings. Incorporating more comprehensive data, such as clinical assessments, laboratory results, and patient-reported outcomes, would strengthen the study and enhance its clinical relevance.

In conclusion, while the manuscript explores an intriguing topic and addresses important concerns, the significant flaws identified throughout the review render it unsuitable for publication. Addressing the lack of relevant references, strengthening the introduction and methods sections, providing logical explanations for the study design, and expanding the dataset to include additional relevant information is essential to improve the manuscript's quality and scientific merit. 

I have carefully reviewed your manuscript, and I suggest some moderate modifications to improve the quality of the writing. Overall, your manuscript is written satisfactorily; however, a few improvements can significantly enhance the clarity and impact of the manuscript.

Author Response

(The authors gave the same response as above.)

Round 2

Reviewer 2 Report

The author answered my question and I am generally satisfied. But in question 3, it appears that аеэевдзпехш до шге асяеьяея It's not English. Also, is there a significant difference in question 6 due to the small SD (average/SD is larger)?

Author Response

Dear reviwer please find the answers to your concerns.

Reviewer 4 Report

Based on the changes I have suggested, incorporating them into the manuscript could greatly improve its chances of acceptance. By addressing the identified issues and implementing the revisions, the manuscript would demonstrate a more rigorous and comprehensive approach, enhancing its overall quality. I recommend the authors consider incorporating these changes, as it would strengthen the manuscript and increase its likelihood of being accepted.

Comments-

1- Please review and correct Figure Number 3, as the patient number appears to be mixed with the description. It should be positioned above the description. Kindly double-check and rectify it accordingly. 

Line - 205- We can assume that the long-lasting therapy is influencing negatively the adherence.

Again, I need clarification on this sentence. In scientific writing, presenting findings as concrete facts is important rather than making assumptions. Instead of using the phrase "we can assume," it is advisable to provide a clear and concise explanation of the significance of the study and its results. By emphasizing the implications and relevance of the findings, readers will better understand the study's contributions to the field. Please revise the sentence to express the significance of the research and the results with greater clarity and avoid using speculative language or assumptions.

In line 271, the authors stated, "We can argue that older patients have worse adherence to therapy." This statement requires further clarification. It is unclear whether the authors are uncertain about this claim. It is crucial to provide a more detailed explanation and establish a clear connection to the findings presented in this study. Please elaborate on this statement, considering the relevant results and their implications.

Discussion section-

To ensure a more comprehensive discussion, it is recommended to provide a more pronounced and detailed background of the topic before directly stating and explaining the results. By setting the context and establishing the groundwork for the discussion, the readers will better understand the significance and relevance of the findings. Therefore, please consider initiating the discussion section by offering a thorough background of the topic, outlining key concepts, previous research, and any relevant theories or frameworks. This will create a stronger foundation for presenting and interpreting the results more coherently and meaningfully.

Limitations of our study are in the fact that it includes patients from only one clinic, 340 but we must note that the total number of MF patients in Bulgaria is appr. 280 and out of 341 them nearly 70 (25%) are treated at SHATHD in Sofia.

Please re-correct the "appr" to its full form in this line. Writing the full expansion is required to present any scientific manuscript. 

Conclusion section- Expanding the conclusion section would be beneficial for effectively presenting the manuscript. While it is unnecessary to provide a complete reiteration of the entire manuscript, offering a more comprehensive discussion in conclusion, including the study's rationale, key outcomes, and the potential future implications of the results, would significantly enhance the overall quality and impact of the paper.

Author Response

Dear Reviwer,

please find the answers to your suggestions. Thank you. 
